# STRUCTURE-RICH TEXT BENCHMARK FOR KNOWLEDGE INFERENCE EVALUATION

## ABSTRACT

We construct a benchmark for LLMs (Large Language Models) composed of structure-rich and syntactically rigorous corpus with semantics-independent tasks, in purpose of evaluating the abilities of knowledge inference from small structured text and construction rules. The tasks also involve the capacity to generate strictly formatted response given the specification, i.e. to output the same structure-rich texts as the inputs. We also experimented on the popular LLMs with our benchmark to compare their competence to mine for information from syntax and condense information into structure.

## 1 INTRODUCTION

### 1.1 BACKGROUND

The explosive growth in the digitization of information in the world today has seen an evolving shift in the dynamics of textual data presentation. Textual communication, encapsulated traditionally in natural human languages, has diversified into more structured formats for seamless interactions among software programs. Notable examples include JavaScript Object Notation (JSON), Python, C++, Yet Another Markup Language (YAML), among others. These structured texts serve as a critical medium in various sectors like technology, health, engineering, finance, where they have found widespread application in data representation, configuration files, and codebases.

Language models, the brain behind most AI technologies, develop a comprehensive understanding of these structured formats. The ability for an artificial intelligence model to understand and generate these structured texts, therefore, presents a significant stride in contemporary global digitalization efforts. This understanding equates to unlocking a new level of competence and functionality in language models. It promises better system interoperability, more efficient parsing and interpreting of structured data, and an eventual enhancement in machine-learning-based applications. It also has far-reaching implications on the development of more adaptive and complex AI systems. Therefore, bringing language models to understand structured texts is not just a necessity, but a fundamental requirement in fast-tracking our collective global digitalization strides.

### 1.2 MOTIVATION

Current attempts to bridge the gap between language models and understanding of these structured texts have, however, met several limitations. Conventionally, language models have predominantly focused on understanding natural human languages, with limited attention to structured texts like JSON, YAML, and codebases. This insufficient attention can be attributed to the growing complexity, diversity, and inherent characteristics of those structured languages, such as abstract data types like trees and graphs. In recent years, significant efforts have centered on broadening the understanding capabilities of language models, with the existing body of work demonstrating considerable advancements in this regard.

However, these studies often operate within a narrow focus, primarily exploring the capacity of these models in generating and interpreting codes—a crucial aspect of structured text understanding. Although these approaches have generated promising results, their limited scope has left a critical part of the solution space unexplored. They largely concentrate on analyzing language models' abilities to decipher programming languages like Python and C++, leaving out other forms of structured

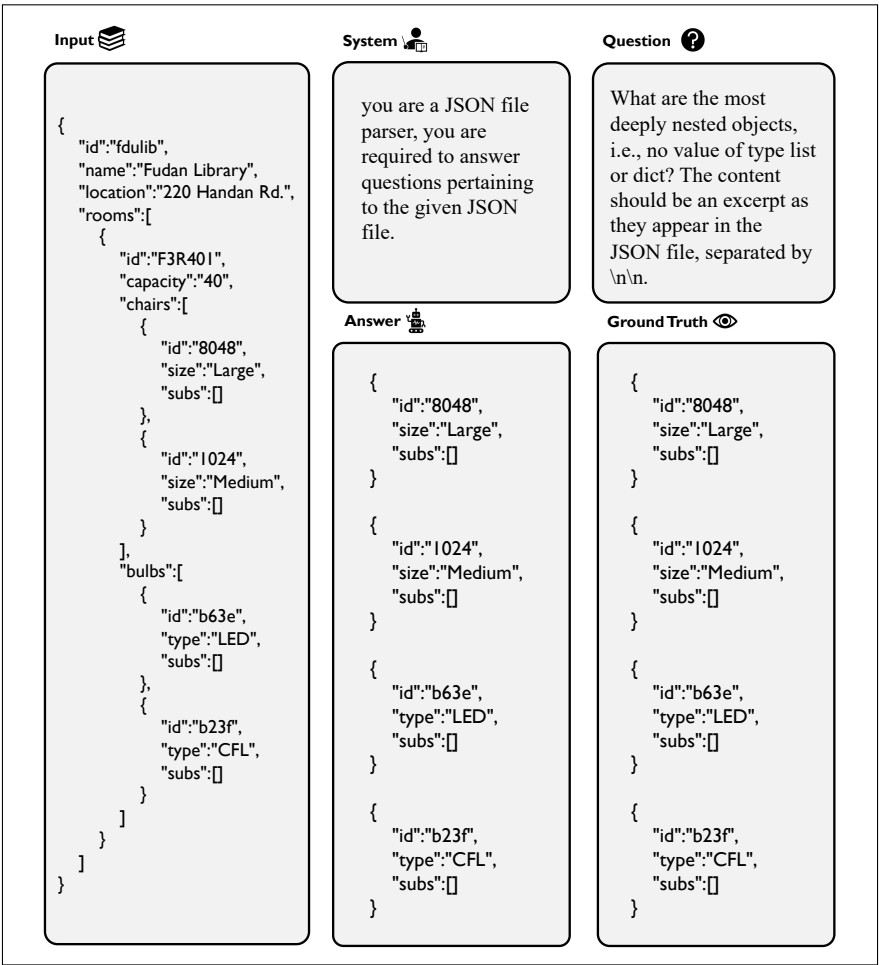

Figure 1: Sample input and tasks of tree.

texts, most notably JSON and YAML. Moreover, these current works do not sufficiently account for the understanding of intricate data structures (like trees and graphs) inherent in structured texts—an essential competency for comprehensive textual understanding. This insufficiency in existing studies signals a compelling need to embark on further explorations. It exposes the urgency to develop a more holistic approach predicated not just on understanding programming codes but on a broader spectrum of structured texts.

### 1.3 OUR PROPOSAL

In this paper, we propose a novel benchmark for testing language models' ability to understand and generate different types of structured texts, like JSON, YAML, and programming languages like Python.

### 1.4 OUR METHOD

We searched vastly structured texts that have well-documented syntax and are widely used in industry and academies. We include both fully-structured and half-structured data to enrich the diversity of our dataset. For each class of texts, we designed 3-5 tasks that are structure-related, some of which are specific to their classification.

Based on the syntax and rules for construction, we randomly generate the input texts, and procedurally generate the corresponding tasks data in the meantime. For some classes whose texts are easily

accessible, we collect the texts from Internet at first and make use of the available parsers to analyze and extract the structure information that constitute the tasks to be constructed.

### 1.5 CONTRIBUTION

- We first thoroughly evaluated the LLMs' ability to understand and manipulate structure-rich texts.

- We present a taxonomy for structured texts and designed structure-related tasks for each class.

- With a total of 9 text classes and several tasks for each class, we present 32 structure-specific tasks and 2512 samples for our benchmark dataset.

- We also tested 4 LLMs and present the results under various metrics for assessment.

The remainder of this paper is arranged into 6 section. section 2 covers attempts in literature to build better models for structural texts centered tasks and construct benchmarks and datasets for such tasks, which have both focused on codebases and fell short of the diversity of structure-rich texts. In section 3 we propose a taxonomy for structure-rich texts that are popular in LLMs' use scenarios and define their generation rules we used, common applications and tasks we designed for each class. section 4 covers our method and framework for collecting input texts and QAs for out dataset. section 5 shows our evaluation on the benchmark for 4 LLMs, namely, GPT-4, Spark, Minimax and Ernie. In section 6 we discussed the performance (scores) distributed among the LLMs and different input types. We wrap up our work and experiment results in section 7.

## 2 RELATED WORKS

Previous explorations over structural information in text have mainly focused on programming language. Due to the difference between normal natural language texts and rich-structure texts, NLP models trained on natural language perform worse than models curated for structural tasksMou et al. (2016). Mou et al. (2016) built a tree-based convolutional neural network that outperformed NLP models on functionality classification tasks and pattern detection tasks over programming languages. Wang et al. (2020) achieved SOTA on code clone detection using graph neural networks and AST representation augmented by control flow of programming language inputs, which is different in architecture from common language models for NLP tasks. Zhou et al. (2019) built graph neural network to perform classification over learned rich node representations gained considerably increased accuracy and F1 scores. The weakness on structural text processing and achievements from alternative architecture shows insufficiency of attention on the realm of rich-structure text from the NLP models.

As for benchmark and datasets, previous works focus on codebases and scenarios of clone detection, vulnerability identification and so on. Svajlenko et al. (2014) proposed a benchmark over Java codes targeting on code clone detection tasks, leaving the inner structure of codes texts untouched. Zhou et al. (2019) also built a dataset from diversified large-scale open-source C projects (Linus, QEMU, Wireshark, FFmpeg) with a total of 48687 commits, manually labeled as vulnerable or non-vulnerable used for vulnerable function identification. The CODESEARCHNETHusain et al. (2019) for code query tasks consist of 6 million functions from open-source code, covering six programming language. It is noteworthy that the construction of the dataset leveraged the information in documentation associated with functions during data filtering. Our work also leveraged the information beyond code texts of which documentation and comments are most typical, when generating the ground truth for one of the python code tasks. Despite that Husain et al. (2019) used it to collect model input instead of ground truth, the similarity of the intention lies in the utilization of the semantic information as much as possible. However, rather than inspecting syntactic and structural information, the tasks in Husain et al. (2019) are semantic code search, which is more semantic-dominated. Other code related datasets include Raychev et al. (2016b) which contains 150000 JavaScript files with parsed ASTs, and Raychev et al. (2016a) which is the Python counterparts to the former one.

## 3 TAXONOMY

### 3.1 INFORMATION BEHIND STRUCTURE

Previous benchmarks on LLMs have largely relied on the tasks where semantic information is critical and central to the correct response, e.g., question answering, summarization and information retrieval. Such tasks cover the majority of usage scenarios for language models, in which inputs are mainly sequential. However, they fail to demonstrate the ability of understanding and manipulating the structure of texts, which is too ubiquitous to be noticed in our daily use of texts.

The commonly used structure-rich texts are normally concise in volume. But through construction rules, they convey ample information that is not explicitly present in the input texts, demonstrating their expressiveness. Those implicit knowledges are inferred on-the-fly according to the rules and structure, starting from the source texts. The rules or syntaxes serve as a function that procedurally leads to different answers under different inputs(Such expressiveness is also characteristic of functional programming where lazy-evaluation is the main trait).

The implicit information is dark, in the sense that it is not readily present or detectable in the given input texts, but it dominates the construction and formation of texts (it is also massive in volume, if explicitly elicited).

Based on the expressiveness and information obscurity of structure-rich texts. We conclude that the abilities to understand these texts and tackle relevant tasks are critical to LLMs that excel in inference and generalization.

### 3.2 TAXONOMY: STRUCTURED-RICH TEXT AND TASK

We searched present literature broadly and propose a overarching taxonomy that covers widely used structure-rich texts, including structured texts (table in relational database), semi-structured texts (XML, JSON, YAML, etc.), abstract data structure in ascii (Tree), and programming language (Python). For every class of input texts, we designed 3-5 tasks regarding the structure and semantics of the input texts. Most tasks are strongly correlated with the structure of input and their ground truths can be derived using program. On the other hand, a few tasks are semantics dependent, mostly when understanding the task instruction. Amongst all 32 tasks, only one task has answer that can not be procedurally obtained from input text. The examples of the input texts and tasks for each category are documented in A.4. For each task, we generated 20 sample input and constructed a dataset of 2512 QAs in total. In this subsection we will introduce each format class as perspective to their definition, application and tasks designed specifically. For a more detailed constructing rules we used to generate input data, see A.1. For example data for each category, see A.4.

### 3.2.1 ABSTRACT DATA STRUCTURE

In the realm of digital information, data is organized in a structure that could easily be stored, parsed, analyzed by machines. The theoretical model underlying these structures are abstract data structures, e.g., tree, list, graph, queue, stack, etc. Abstract data structure decides how data is queried, inserted, modified and deleted. These structures encode the relations and properties of data. The ability to understand abstract data structure will enable LLMs to rigidly follow the relation of semantic segments imposed from syntax, which is vital to data exchange between different format. The ability of LLMs solving graph problems has been explored in previous work (Wang et al. (2023)).

Tree, from the view of graph, is acyclic connected undirected graph. It is defined in the same way as a graph, i.e., a set of vertices and a set of edges connecting vertices thereof. Given the pervasive influence of tree on other recursive structured texts such as JSON, YAML and XML, we focused on tree structure and designed tree-specific tasks.

Regarding each tree, we designed 3 tasks pertaining to their structures, all of which are independent of the content in the nodes. A sample of input texts of tree category along with its associated tasks are shown in Figure 5. These tasks target on the ability to compose a path to a specified node and decide the height or depth of a node.

### 3.2.2 STRUCTURED

Generally, structured data refer to those well-structured data that obey certain elaborate data model which defines the relation of data elements and abstract models. Tabular data exported from data tables in relational databases are typical of such structured data. Data that are structured according to non-tabular data models are also deemed as structured data.

In this work, we focus on specific row-column structured text inputs. Such input denotes a set of objects with the same set of fields defined by the first row. The subsequent rows denotes the values of fields per object. The format is derived from tables in relational database, which has seen overwhelming domination in the realm of database.

Due to the simplicity of the input, apart from a value lookup task, we built statistical tasks as well as multiple-table tasks that are similar to inner join query, to better capture the features and practical use scenarios of such input data.

### 3.2.3 SEMI-STRUCTURED

As opposed to structured texts, semi-structured data are structured data that do not fit in the conventional data model (Buneman (1997)). Semi-structured data are structured and contain rich structural marks or tags that are different from semantic elements and compose the structure they are defined with. In this work, we focus specifically on texts with some extent of hierarchies designated by marks and reserved tokens. They are neither sequential nor tabular, contrary to row-column structured table as in subsubsection 3.2.2. Internally they are n-ary trees, which are dominated by recursive construction rules. Semi-structured data are employed in information exchange between application programming interfaces (JSON, YAML), storing well organized web pages on world wide webs (XML) and typesetting journals or scientific articles (Markdown, Org, LaTeX). All of these medium have been backbone of our digitalized world. In our effort to define subsets of these formats (see A.1) and generate some sample input texts from the rules we found that a simple pattern kept recurring in their construction. The pattern that intersects with every semi-structured text type we covered here is recursion. Each type have some syntax that is simple and atomic, which contribute little to the hierarchy or structural complexity, along with optional recursion, which is the essence of hierarchy construction and non-sequential structure.

#### OBJECT NOTATION

Object Notation is a format that could be used in modeling the in-memory objects data conceptualized in object-oriented programming language.

**JSON**

JSON is acronym for JavaScript Object Notation. It has been widely used in data storage, exchange, transmission and inter-operation between different API. Main stream programming languages offer complete support for exporting and importing JSON files, making it popular on occasions of complicate structured data transmission. Therefore, LLMs having a better understanding and manipulating ability with respect to JSON texts input will expedite the progress of LLMs bridging the gap between different participants and procedures in digital data flow happened in real world.

Due to the inherit hierarchy structure of Object Notations, we adopted a recursive scheme to define our input texts (see A.1.3). For JSON text, we built five tasks, inspecting abilities covering information retrieval, structure traversal, path construction, syntax correction, and depth calculation.

It is noteworthy that the depth calculation task is in essential a task of tree structure, which is in alignment with tree category in subsubsection 3.2.1. Another insights rooted in the fact that we used the same tasks interchangeably with little modification to the generating code for JSON and YAML, demonstrating their similarity internally.

**YAML**

YAML (Yet Another Markup Language or YAML Ain't Markup Language) is a data-oriented markup language that models object data. It supports a compact in-line style (although not recommended) that is equivalent in semantics to JSON. The difference lies in the additional syntax that

JSON does not support, e.g., comments. Besides, YAML is widely used in configuration files other than object serialization.

**XML**

XML(extensible markup language) is a markup language, but it is also widely used in configuration files and storing object and metadata. Here we focus mainly on its usage in object notation, instead of marking up documents.

Its syntax defines the specifications for markup (tags are typical markups in XML) and contents and recursive construction of elements constituted by tags, attributes and contents which could contain nested elements. We designed tasks regarding syntax correction, tag-based text retrieval and attribute-based text retrieval.

In terms of the influence in digitalization and importance of LLMs mastering YAML format, the same trend applies to YAML. In addition, XML as the foundation of world wide web data, mastering XML well will benefit LLMs in taking advantages of resources on the Internet better and promote the level of intelligence of language model.

### MARKUP LANGUAGE

The syntax rules in markup languages are used mainly for markup excerpts of texts rather than modeling entities as others we mentioned above. Despite that the syntax of ORG, Markdown and LaTeX are no less complicated than Object Notation languages as of Chomsky hierarchy (Chomsky & Schützenberger (1959); Schützenberger (1963)), their common use scenarios are less dominated by recursive structure objects. So our tasks focus more on text retrieval and shallow-nested structure traversal. Nevertheless, we built longer input text to diversify our dataset.

Our construction of these formats used a small subset of legal commands with limited depth in nested (sub)sections.

**Markdown**

Markdown is a lightweight markup language for formatting text using reserved characters scattered among the content. The main goal of Markdown is to format text in a source-readable way. It has gain great popularity in technical writing, blogging, and collaborative documentations despite the syntax standardization is underway still and implementation is diverged.

**Org**

Org files are plain text files that include simple plain-text marks to express hierarchy. It is easy to read (for most of the time, but it could be intricate too) in source code form, which is similar to markdown. the syntax is intended to be read and modified by editors in a automated manner, which is richer and more customizable than markdown.

It is designed specifically for Org-mode of Emacs editor. The editor can read the markups in Org files and manipulate the hierarchy and other markups as a result of toggling outline objects or element status, e.g., check off to-do list items with a simple keystroke under Org-mode in Emacs. Note-taking and list management is Org's forte.

**LaTeX**

LaTeX is a set of TeX macros that has gain great popularity. It has simple syntax and rich command set, being highly extensible. It has been heavily used in academia for typesetting scientific documents.

Contrary to markdown and org, LaTeX file is hard to read and much more verbose. It has the most features in all markup languages we covered as a full-fledged typesetting tool.

### 3.2.4 PROGRAMMING LANGUAGE

Python is a dynamically typed language that has seen exclusive application in mathematics and data science. It has rich features that make it procedural, object-oriented and functional at the same time. The main trait of Python is usability (garbage collection provided by language), and expressiveness (it has syntax that captures most frequently recurring programming patterns, e.g., generators).

Due to its complexity in structure and syntax, we collected from Internet [1] 1932 files instead of procedural generation, and used ast module [2] and ast_scope module [3] to parse the source codes into abstract syntax tree. We designed problems regarding identifier scope and return type (when it is annotated explicitly). The answers are derived from parser, i.e., static analysis. We also have a task querying the purpose of the code, i.e., algorithm or model the code implements. For answering such tasks, we used the information in the python file names (out of input texts) to get the answer. This is the only task that require the understanding of input file semantics and can not be solved in a systematic AI way.

# 4 DATA CONSTRUCTION

Based on the construction rules in A.1, we procedurally generated input texts with randomly generated placeholder content. And during the procedure of generating input texts, we have collected informations regarding outline hierarchy, identifiers used in markup or syntax marks, statistical properties and mapping from identifier to contents. With these information, we can find the ground truth to our designed tasks systematically. For the syntax correcting tasks, we will incorporate intentional syntax error with a probability, and the input texts for such tasks are constructed individually.

Among all the file format we covered, python input texts are not procedurally generated, but rather collected from Internet, which are rich in semantics with real-world application. The ground truth tasks asking the algorithms or models the codes intend to implement is taken from the filenames of python files which is an indicator of the models or algorithms under the hood.

Randomness is the core feature we tried to maintain during the procedural generation, since we want to mangle the semantics of out input texts and focus solely on structural information. The contents and identifiers used in syntax marks are randomly generated from a finite set of character elements.

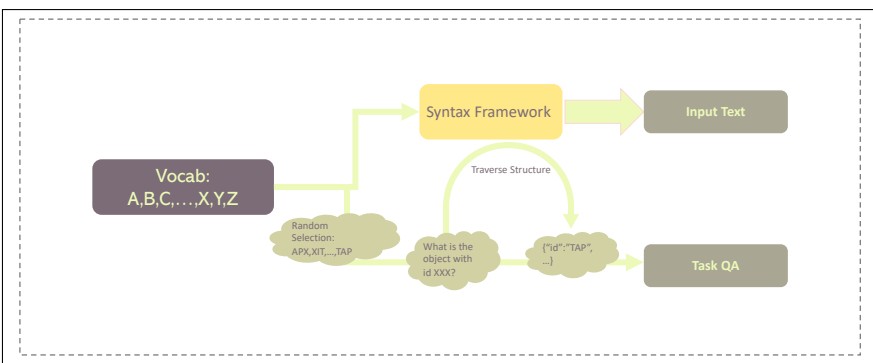

Figure 2: Process of input texts generation

# 5 EVALUATION

With a dataset composed of 2512 QAs, we evaluated on 4 LLMs through network API, and adopted a heuristic elicitation method to offer more hints. To clarify, we queried each LLM with input text concatenated with question, and the response is evaluated against ground truth by exact match, rouge-1 score and T/F result from another judging LLM.

## 5.1 BASELINE

First we used GPT-4 as baseline (due to inaccessibility of GPT-4, the experiment evaluated PYTHON and Org input on GPT3.5), i.e., only provide question with input text, without prompt

---

[1] https://github.com/TheAlgorithms/Python
[2] https://pypi.org/project/AST/
[3] https://pypi.org/project/ast-scope/

engineering and techniques alike. Evaluation of other LLMs are conducted under the same baseline setting, including Minimax [4], Spark from Xunfei [5] and Ernie from Baidu [6].

## 5.2 PROMPT ENGINEERING: HINT ELICITATION

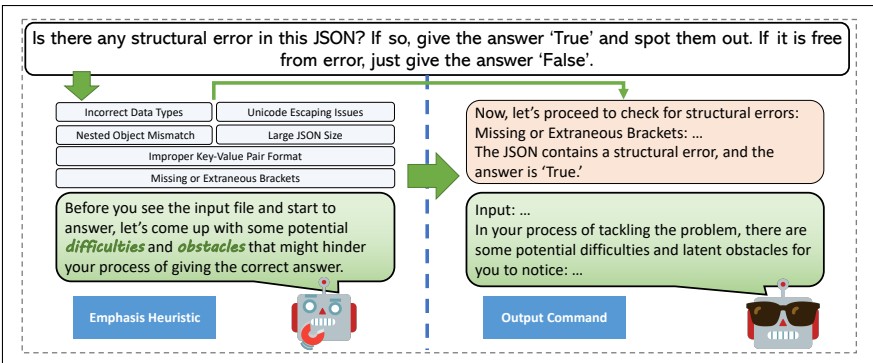

Figure 3: Process of eliciting hint from LLMs to improve prompt

Apart from baseline experiment, we also explored the ability of LLMs to think of hints for task solution. The detailed framework is demonstrated in Figure 3. We use an eliciting query regarding the task's question to have LLMs generate some key points or obstacles before seeing the input texts. After the hints collection step, we will repeat baseline experiment with a difference in attached hints elicited before head, in hope that LLMs tackling these difficulties will help them think more verbosely to avoid errors.

## 5.3 BACKGROUND KNOWLEDGE ENHANCEMENT

The structured text is pretty concise and expressive, most of its information is not readily explicit. There are a large volume of data that need to be inferred combining the raw input and structure information (or syntax rules). That leads to a intuitive conjecture that LLMs will likely to perform better on these structure dominant tasks provided that they have enough background knowledge about the syntax and construction rules for the rich-structure text in concern. To check the correlation of background knowledge on structure and the ability to answer the questions in our dataset, we conducted experiment with background knowledge included in the prompt when querying LLMs.

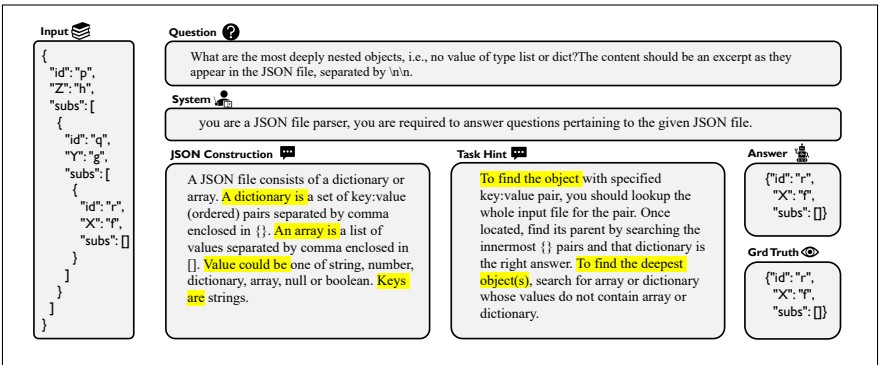

Figure 4: Adding background knowledge for JSON tasks

[4] https://api.minimax.chat/v1/text/chatcompletion_pro

[5] ws://spark-api.xf-yun.com/v1.1/chat

[6] https://aip.baidubce.com/rpc/2.0/ai_custom/v1/wenxinworkshop/chat/eb-instant

## 6 ANALYSIS

### 6.1 BASELINE

Under the baseline experiment setting, GPT-4 outperforms all other LLMs with a significant margin and obtained accuracy over 0.96 in JSON, YAML, XML data, showing its proficiency in text retrieval, structure traversal, path construction, syntax correction, and depth calculation regarding object notation input texts. It answers only around half of the questions about Tree and Org. Tree input texts have the simplest syntax but require considerable calculation to rebuild the tree structure, which could make them difficult tasks for generative architecture. The Org type has similar syntax Markdown but with a relatively narrower niche in application and limited popularity. Such under-exploration means the input texts and its syntax might be entirely strange for LLMs, leading to the lack of necessary knowledge to do the inference towards solution. In terms of Tabular, Markdown, LaTeX and PYTHON, GPT-4 has gain similar scores around 0.7, which reflect the less familiarity over these inputs, given that the tasks are much easier as opposed to object notation input.

The performance of Minimax distribute over 8 types (other than Tree) similarly to GPT-4, except that the respective score is lower by 0.1-0.2. It barely answers correctly on tasks regarding Tree input.

As for Spark, its highest scores are earned in JSON and PYTHON tasks. YAML, XML and Markdown tasks shares scores around 0.5. Ratio of correct answers in Tabular, Latex and Org fell into around 30 percent. The score for Tree is 0.089 and lowest, which is the same as other LLMs.

The Ernie model gains its highest score around 0.6-0.7 in object notation types, with sub-optimum performance on Markdown and PYTHON, around accuracy of 50 percent. Tabular and LaTeX input texts have seen a score around 0.36. On Tree input, it earned its lowest score of 0.133, which is a marginal promotion over Minimax and Spark.

### 6.2 HINT ELICITATION

For all 9 classes, only Tree, tabular and PYTHON has seen enhancement in three LLMs from hint eliciting prompt engineering. For object notation texts, hint has seen lower accuracy, with one exception on Ernie for JSON tasks. For markup languages, hint improved accuracy on Ernie but worsen the outcome on Minimax and Spark. In each cases, the baseline GPT4 possessed the highest scores. A detailed demonstration is listed in A.3

### 6.3 BACKGROUND KNOWLEDGE

For background knowledge experiment, the result (see 8) from testing JSON input texts with background knowledge appended to query shows that such alternation is insufficient to enhance the performance of LLMs. The result could also be attributed to diverse factors.

- The limited understanding on these background knowledge
- The additional background knowledge has been included in the training set of LLMs, making the information superfluous
- The LLMs are currently unable to do complicate structul or logical inference, i.e., they failed to apply given rules on the raw input texts

## 7 CONCLUSION

We presented a taxonomy involving 9 popular structure-rich texts, with diversity in structural complexity and domain applied. We classified these inputs into 4 first level category by use scenarios and demonstrated their definition, construction rules we used in generation and common applications. By procedural construction and parsing, we built a benchmark with 2512 QAs and evaluated GPT-4, Minimax, Spark and Ernie. The experiment uncovered the under-exploration of structure-rich input texts for LLMs other than GPT-4, among which Tree is the most typical and where poor performance occurs. Prompt engineering of hint elicitation and additional background knowledge have both failed to address such inefficiency well.

AUTHOR CONTRIBUTIONS

XXX

ACKNOWLEDGMENTS

XXX

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

## A  APPENDIX

### A.1  GENERATION OF INPUT TEXTS

#### A.1.1  TREE

We build tree-structured input as a list of edges in a tree, in a format of "`father->child`", separated by newline.

$$identifier := \texttt{[a-z]+}$$
$$Edge := identifier \texttt{->} identifier$$
$$Tree := Edge(\texttt{\textbackslash n}Edge)\star$$

### A.1.2 TABULAR

Formally, input texts are classified as structured texts given that they are composed of a list of newline separated lines, each of which is a list of text cells delimited by comma.

$$
\begin{aligned}
head &\coloneqq \texttt{[A-Z][a-z]}\star \\
cell &\coloneqq \texttt{[A-Za-z0-9]+} \\
headline &\coloneqq identifier(,identifier)\star \\
subline &\coloneqq cell(,cell)\star \\
Tabular &\coloneqq headline(\texttt{\textbackslash n}subline)+
\end{aligned}
$$

### A.1.3 JSON

Due to the inherit hierarchy structure of Object Notations, we adopted a recursive scheme to define our input texts.

$$
\begin{aligned}
lb_{(left\ bracket)} &\coloneqq \texttt{[[]} \\
rb &\coloneqq \texttt{[]]} \\
val &\coloneqq \texttt{[a-z]+} \\
key &\coloneqq \texttt{[A-Z]+} \\
JSON &\coloneqq \{ \\
&\quad \texttt{"id":"}val\texttt{"} \\
&\quad \texttt{"subs":}lbrb|lbJSON(,\texttt{\textbackslash n}JSON)\star rb \\
&\quad (\texttt{"}key\texttt{":"}val\texttt{"}\texttt{\textbackslash n})+ \\
&\quad \}
\end{aligned}
$$

### A.1.4 YAML

The rules for constructing YAML and XML input are similarly recursive.

$$
\begin{aligned}
YAML &\coloneqq \\
&\quad \texttt{id}:val \\
&\quad \texttt{subs}:lbrb|(\texttt{\textbackslash n}(\texttt{\textbackslash t})\star - \ YAML)+ \\
&\quad (key:val\texttt{\textbackslash n})+
\end{aligned}
$$

### A.1.5 XML

$$
\begin{aligned}
firstline &\coloneqq \texttt{<?xml version="1.0" encoding="UTF-8"?>} \\
XML &\coloneqq \\
&\quad firstline \\
&\quad XMLObject \\
tag &\coloneqq \texttt{[A-Z]+} \\
val &\coloneqq \texttt{[a-z]+} \\
attr &\coloneqq \texttt{[A-Z]+="}val\texttt{"} \\
content &\coloneqq \texttt{[a-z \textbackslash n\textbackslash t]}\star \\
XMLObject &\coloneqq \\
&\quad <tag(\ attr)\star> \\
&\quad ((\texttt{\textbackslash t})\star XMLObject)\star \\
&\quad content \\
&\quad </tag>
\end{aligned}
$$

### A.1.6 LaTeX

In LaTeX input texts, we include `textbf` and `includegraphics` commands to accommodate for the text retrieval tasks. The headings serve as anchors for structure traversal.

$$command \coloneqq \backslash(\text{section|subsection|subsubsection})$$
$$heading \coloneqq command\{\text{[a-z]+}\}|\text{[a-z]+}$$
$$inclg \coloneqq$$
$$\backslash\text{includegraphics[width=0.5\textwidth]\{[a-z]+[.](png|jpg|jpeg|gif)\}}$$
$$bf \coloneqq \backslash\text{textbf\{[a-z ]+\}}$$
$$content \coloneqq (\text{[a-z ]}|bf|inclg)+$$
$$LaTeX \coloneqq heading\backslash ncontent(\backslash nLaTeX)*$$

### A.1.7 MARKDOWN

In markdown input texts, the syntax counterparts for heading, text face and including figure are employed in our dataset.

$$heading \coloneqq \text{[\#]} * \text{[a-z]+}$$
$$inclg \coloneqq !lb\text{alt}rb\backslash(\text{[a-z]+[.](png|jpg|jpeg|gif)} \text{ "hover text"}\backslash)$$
$$bf \coloneqq \text{[*]\{2\}[a-z ]+[*]\{2\}}$$
$$content \coloneqq (\text{[a-z ]}|bf|inclg)+$$
$$Markdown \coloneqq heading\backslash ncontent(\backslash nMarkdown)*$$

### A.1.8 ORG

In Org input texts, the syntax is obtained from JSON construction rules by replacing the markups for heading, including figures and bold font face.

$$heading \coloneqq \text{[*]} * \text{[a-z]+}$$
$$inclg \coloneqq lb\{2\}\text{[a-z]+[.](png|jpg|jpeg|gif)}rb\{2\}$$
$$bf \coloneqq \text{[*][a-z ]+[*]}$$
$$content \coloneqq (\text{[a-z ]}|bf|inclg)+$$
$$Org \coloneqq heading\backslash ncontent(\backslash nOrg)*$$

### A.1.9 PYTHON

Python input texts is the only type of texts that is not randomly generated but rather collected from Internet, so the code text should conform with the python programming language syntax as documented in their websites `https://docs.python.org`.

### A.2 EXPERIMENTAL DATA

Table 1: GPT-4 w/o prompt engineering

| Metric | Tree | Tabular | JSON | YAML | XML | Markdown | Org | LaTeX | PYTHON |
|---|---|---|---|---|---|---|---|---|---|
| Exact Match | 0.52 | 0.78 | 0.64 | 0.93 | 0.9 | 0.36 | 0.25 | 0.38 | 0.259 |
| Rouge-1 | 0.107 | 0.107 | 0.383 | 0.368 | 0.195 | 0.894 | 0.496 | 0.944 | 0.068 |
| GPTJudge | 0.556 | 0.7 | 0.983 | 0.967 | 0.972 | 0.736 | 0.5 | 0.819 | 0.787 |

Table 2: Miminax w/o prompt engineering

| Metric | Tree | Tabular | JSON | YAML | XML | Markdown | Org | LaTeX | PYTHON |
|---|---|---|---|---|---|---|---|---|---|
| Exact Match | 0.133 | 0.738 | 0.470 | 0.430 | 0.567 | 0.283 | 0.233 | 0.267 | 0.201 |
| Rouge-1 | 0.101 | 0.105 | 0.501 | 0.490 | 0.206 | 0.690 | 0.506 | 0.742 | 0.091 |
| GPTJudge | 0.067 | 0.525 | 0.850 | 0.770 | 0.883 | 0.683 | 0.450 | 0.683 | 0.679 |

Table 3: Spark w/o prompt engineering

| Metric | Tree | Tabular | JSON | YAML | XML | Markdown | Org | LaTeX | PYTHON |
|---|---|---|---|---|---|---|---|---|---|
| Exact Match | 0.383 | 0.975 | 0.310 | 0.540 | 0.433 | 0.333 | 0.250 | 0.267 | 0.263 |
| Rouge-1 | 0.007 | 0.001 | 0.023 | 0.011 | 0.018 | 0.026 | 0.032 | 0.035 | 0.000 |
| GPTJudge | 0.089 | 0.280 | 0.800 | 0.472 | 0.533 | 0.529 | 0.300 | 0.296 | 0.702 |

Table 4: Ernie w/o prompt engineering

| Metric | Tree | Tabular | JSON | YAML | XML | Markdown | Org | LaTeX | PYTHON |
|---|---|---|---|---|---|---|---|---|---|
| Exact Match | 0.167 | 0.637 | 0.200 | 0.240 | 0.200 | 0.217 | 0.133 | 0.117 | 0.295 |
| Rouge-1 | 0.056 | 0.051 | 0.161 | 0.162 | 0.084 | 0.226 | 0.137 | 0.283 | 0.066 |
| GPTJudge | 0.133 | 0.375 | 0.700 | 0.660 | 0.617 | 0.433 | 0.200 | 0.350 | 0.550 |

Table 5: Miminax w/ prompt engineering

| Metric | Tree | Tabular | JSON | YAML | XML | Markdown | Org | LaTeX | PYTHON |
|---|---|---|---|---|---|---|---|---|---|
| Exact Match | 0.150 | 0.738 | 0.390 | 0.400 | 0.517 | 0.250 | 0.183 | 0.233 | 0.205 |
| Rouge-1 | 0.027 | 0.065 | 0.241 | 0.290 | 0.181 | 0.429 | 0.360 | 0.571 | 0.093 |
| GPTJudge | 0.183 | 0.575 | 0.850 | 0.730 | 0.850 | 0.600 | 0.433 | 0.683 | 0.708 |

Table 6: Spark w/ prompt engineering

| Metric | Tree | Tabular | JSON | YAML | XML | Markdown | Org | LaTeX | PYTHON |
|---|---|---|---|---|---|---|---|---|---|
| Exact Match | 0.483 | 0.963 | 0.420 | 0.540 | 0.833 | 0.417 | 0.400 | 0.300 | 0.268 |
| Rouge-1 | 0.002 | 0.001 | 0.003 | 0.004 | 0.006 | 0.029 | 0.015 | 0.027 | 0.000 |
| GPTJudge | 0.214 | 0.457 | 0.533 | 0.286 | 0.483 | 0.100 | 0.250 | 0.222 | 0.705 |

Table 7: Ernie w/ prompt engineering

| Metric | Tree | Tabular | JSON | YAML | XML | Markdown | Org | LaTeX | PYTHON |
|---|---|---|---|---|---|---|---|---|---|
| Exact Match | 0.183 | 0.637 | 0.230 | 0.160 | 0.217 | 0.100 | 0.083 | 0.133 | 0.238 |
| Rouge-1 | 0.042 | 0.041 | 0.167 | 0.115 | 0.101 | 0.143 | 0.104 | 0.115 | 0.044 |
| GPTJudge | 0.233 | 0.438 | 0.810 | 0.580 | 0.593 | 0.450 | 0.450 | 0.417 | 0.608 |

Table 8: JSON input with background knowledge

|  | Background Knowledge | Minimax:Base | Minimax:Hint | Ernie:Base | Ernie:Hint |
|---|---|---|---|---|---|
| GPTJudege | w/o | 0.857 | 0.859 | 0.7 | 0.81 |
|  | w/ | 0.82 | 0.85 | 0.75 | 0.8 |
| Exact Match | w/o | 0.469 | 0.394 | 0.2 | 0.23 |
|  | w/ | 0.42 | 0.33 | 0.25 | 0.19 |
| Rouge-1 | w/o | 0.504 | 0.243 | 0.161 | 0.167 |
|  | w/ | 0.524 | 0.223 | 0.146 | 0.173 |

## A.3  COMPARATION ON GPT JUDGED ACCURACY OF LLMS

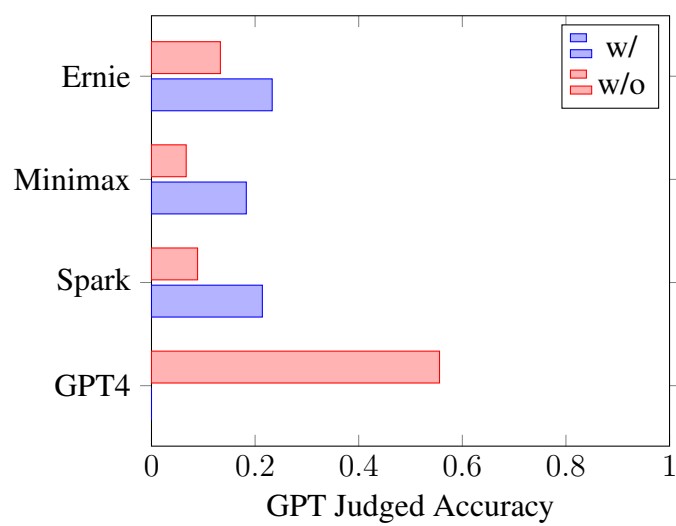

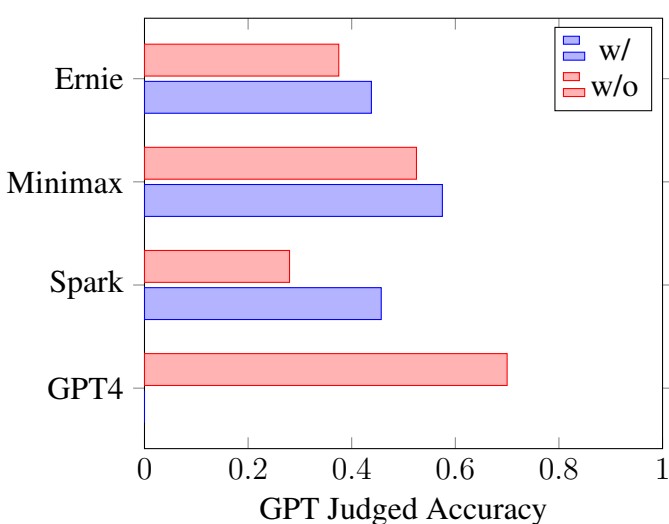

JSON task evaluation based on GPT judge

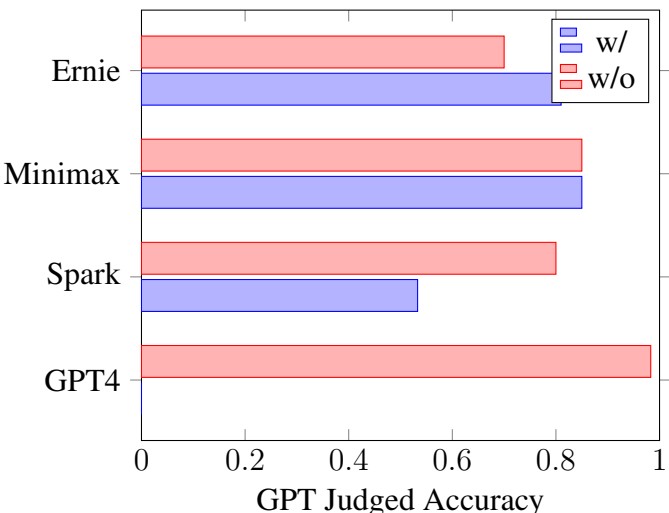

YAML task evaluation based on GPT judge

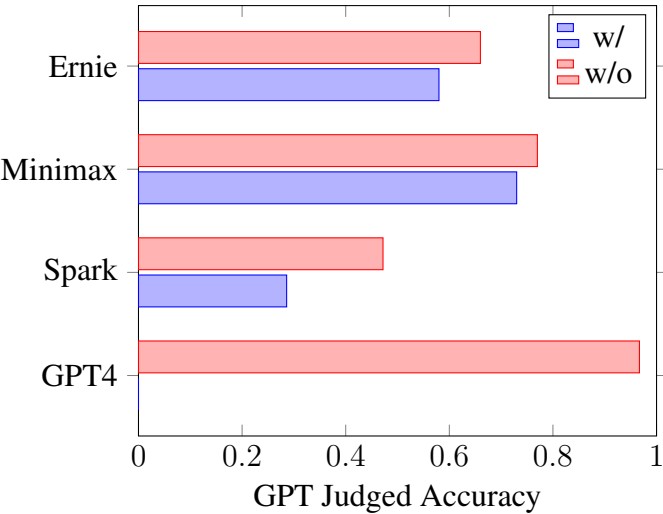

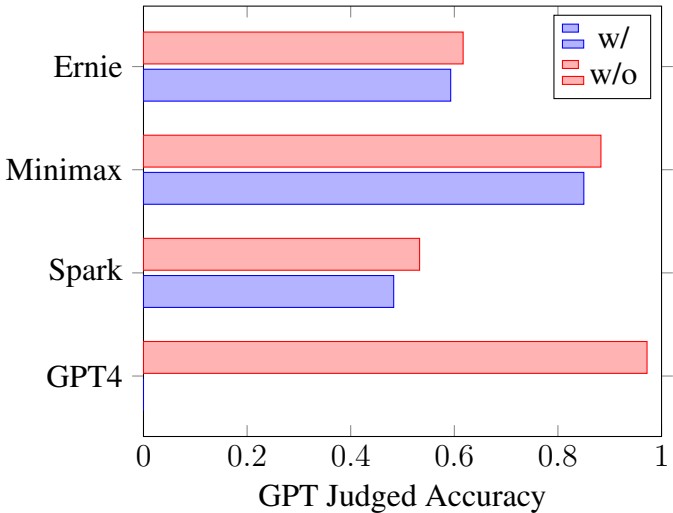

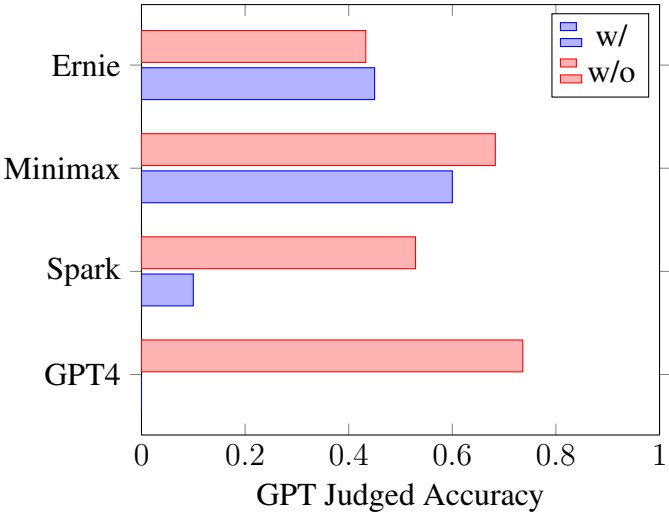

Org task evaluation based on GPT judge

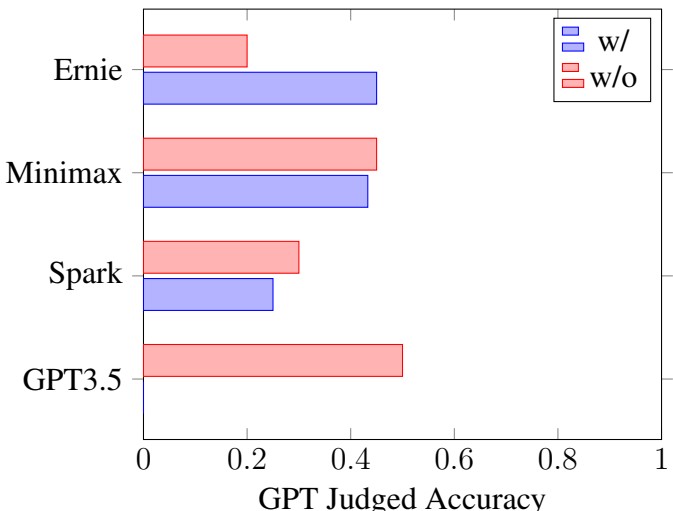

LaTeX task evaluation based on GPT judge

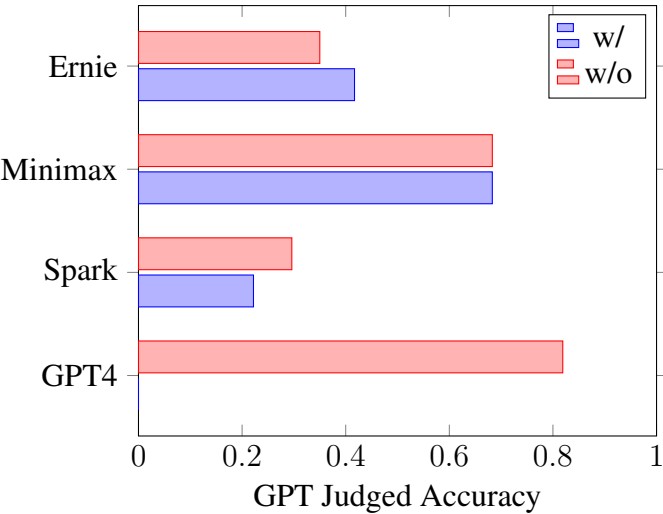

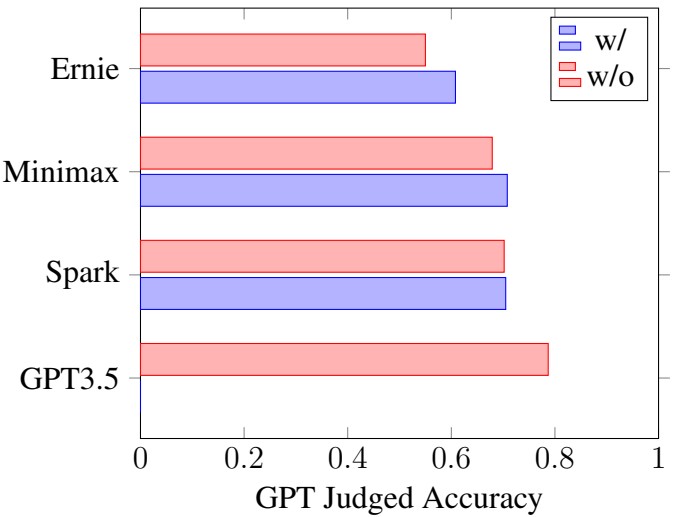

## A.4 SAMPLE OF INPUT AND TASKS

### A.4.1 TREE

See Figure 5.

### A.4.2 TABULAR

See Figure 6.

### A.4.3 JSON

See Figure 7.

### A.4.4 YAML

See Figure 8.

### A.4.5 XML

See Figure 9.

### A.4.6 LaTeX

See Figure 10.

### A.4.7 MARKDOWN

See Figure 11.

### A.4.8 ORG

See Figure 12.

### A.4.9 PYTHON CODE

See Figure 13.

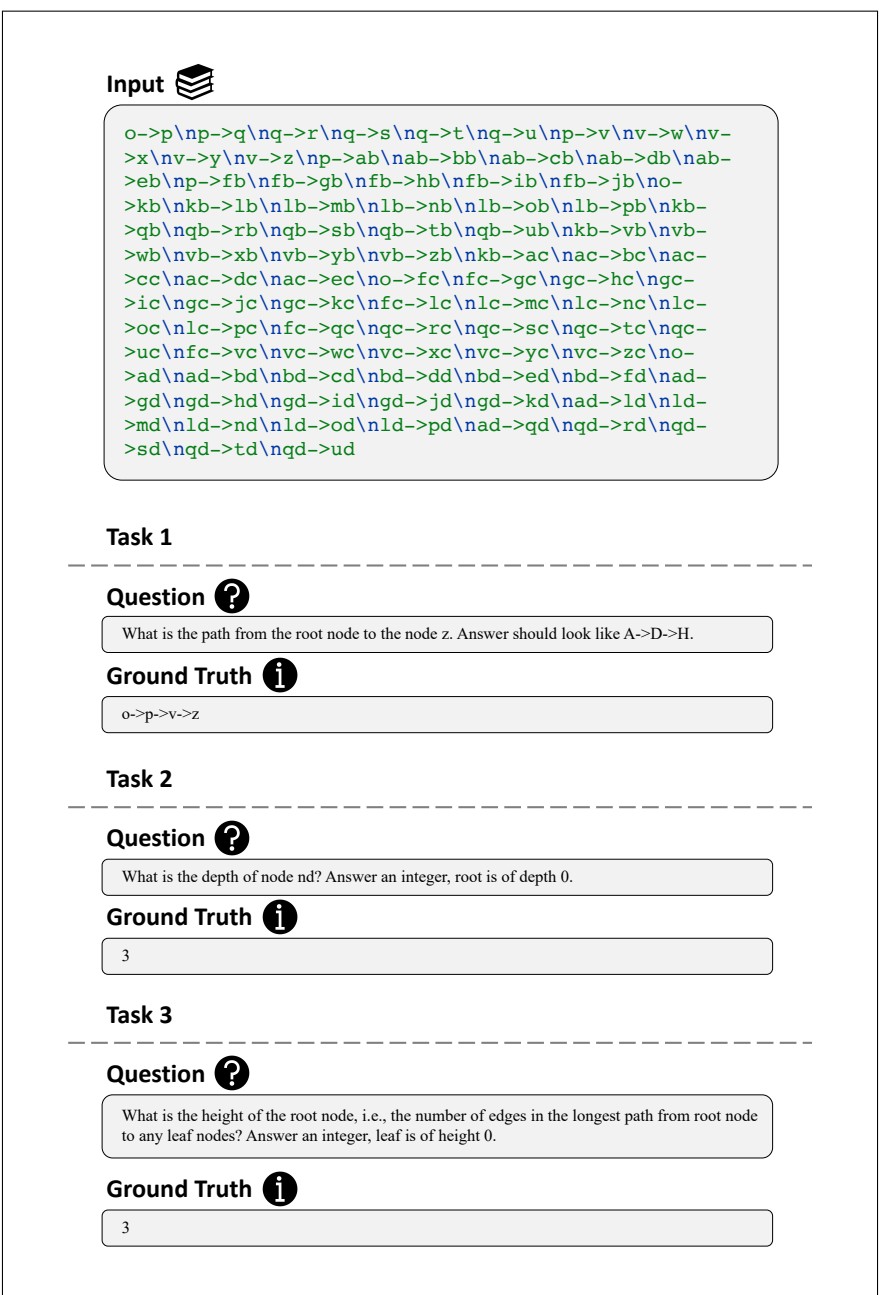

Figure 5: Sample input and tasks of Tree.

**Input** 📚

```
primeKey gender age name  height weight color
a female 23  n 157 144 olive
b male  39  o 191 104 swarthy
c male  14  p 134 162 black
d male  39  q 163 124 brown

primeKey status salary companylocation
a employed 460789 TwitterNY
b retired861910 NVIDIA GA
c retired360565 Meta  CA
d employed 350426 Google GA
```

**Task 1**

**Question** ❓

What is the color of record with primeKey c

**Ground Truth** ℹ️

black

**Task 2**

**Question** ❓

How many people who work in IL are taller than 171?

**Ground Truth** ℹ️

0

**Task 3**

**Question** ❓

How many people work with salary more than 516275?

**Ground Truth** ℹ️

1

**Task 4**

**Question** ❓

How many people are female?

**Ground Truth** ℹ️

1

Figure 6: Sample input and tasks of tabular data.

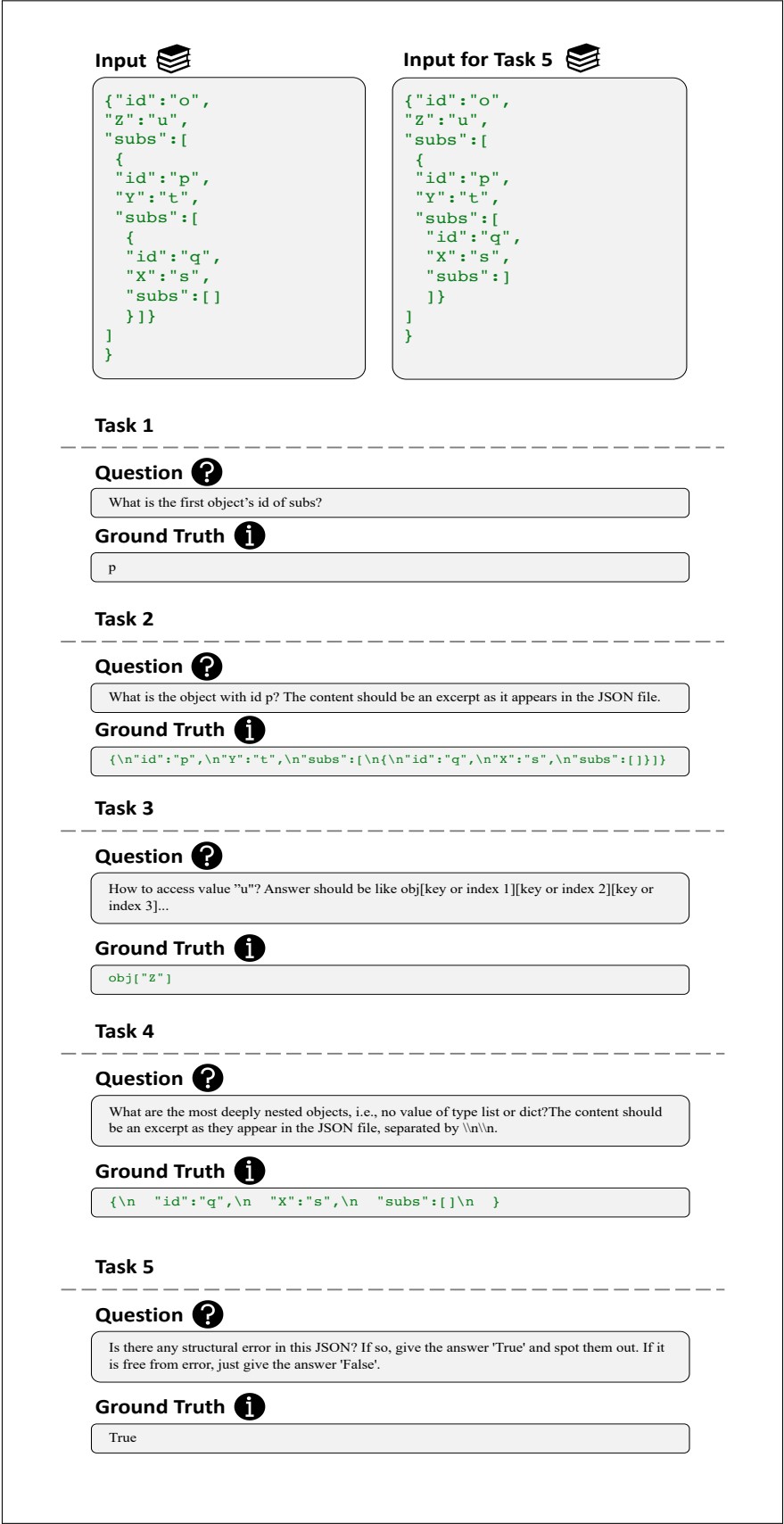

Figure 7: Sample input and tasks of JSON.

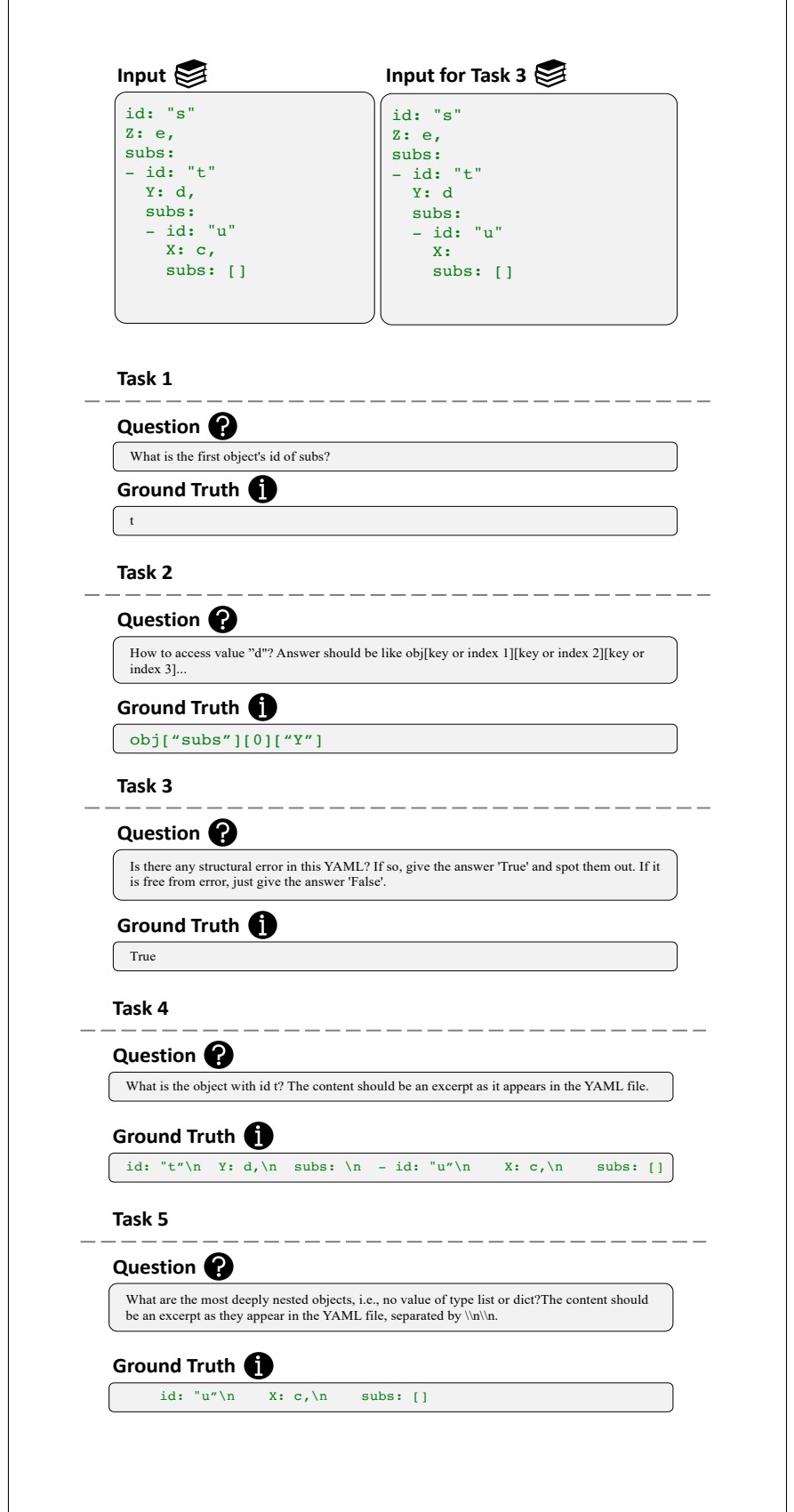

Figure 8: Sample input and tasks of YAML.

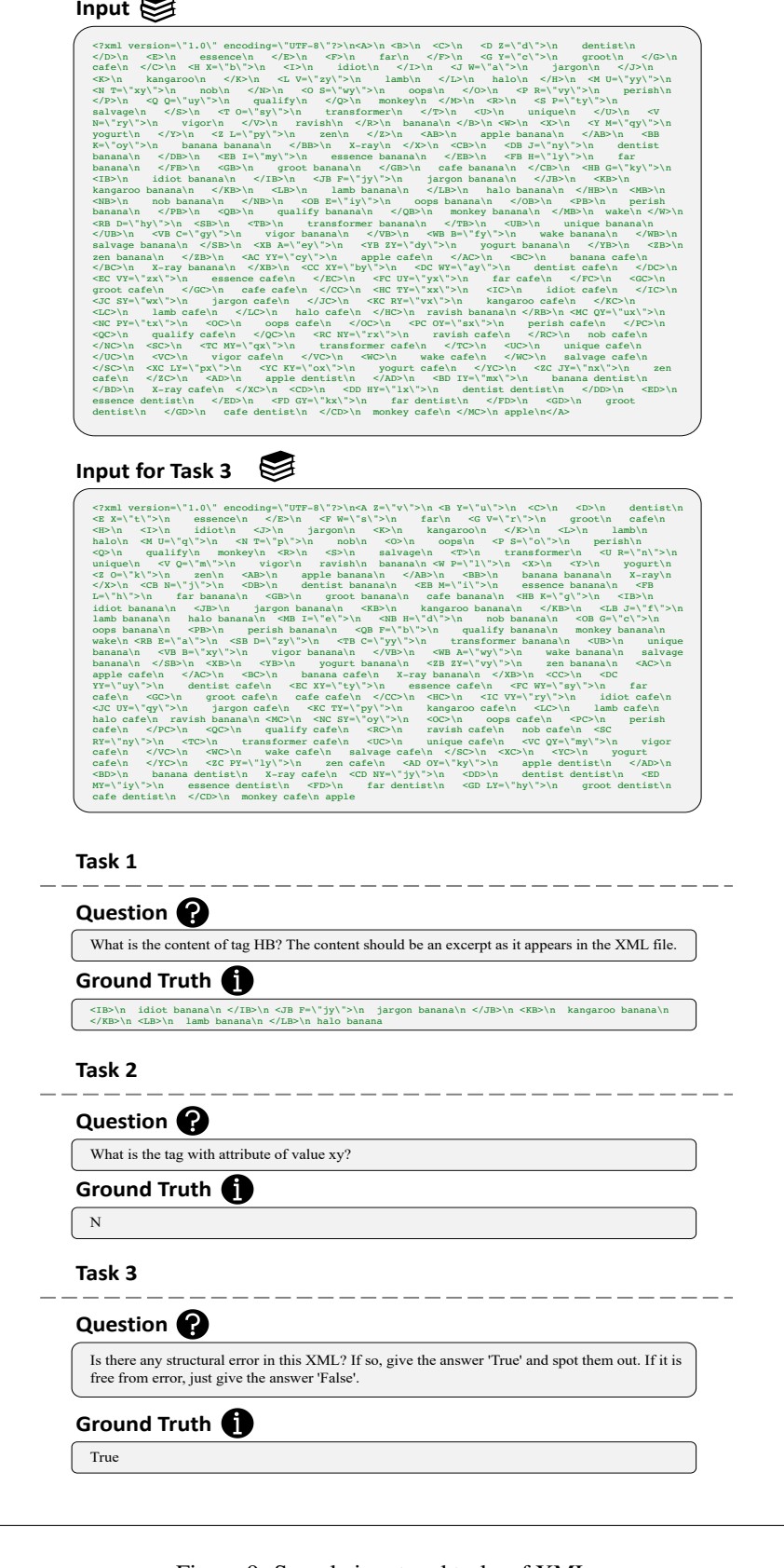

Figure 9: Sample input and tasks of XML.

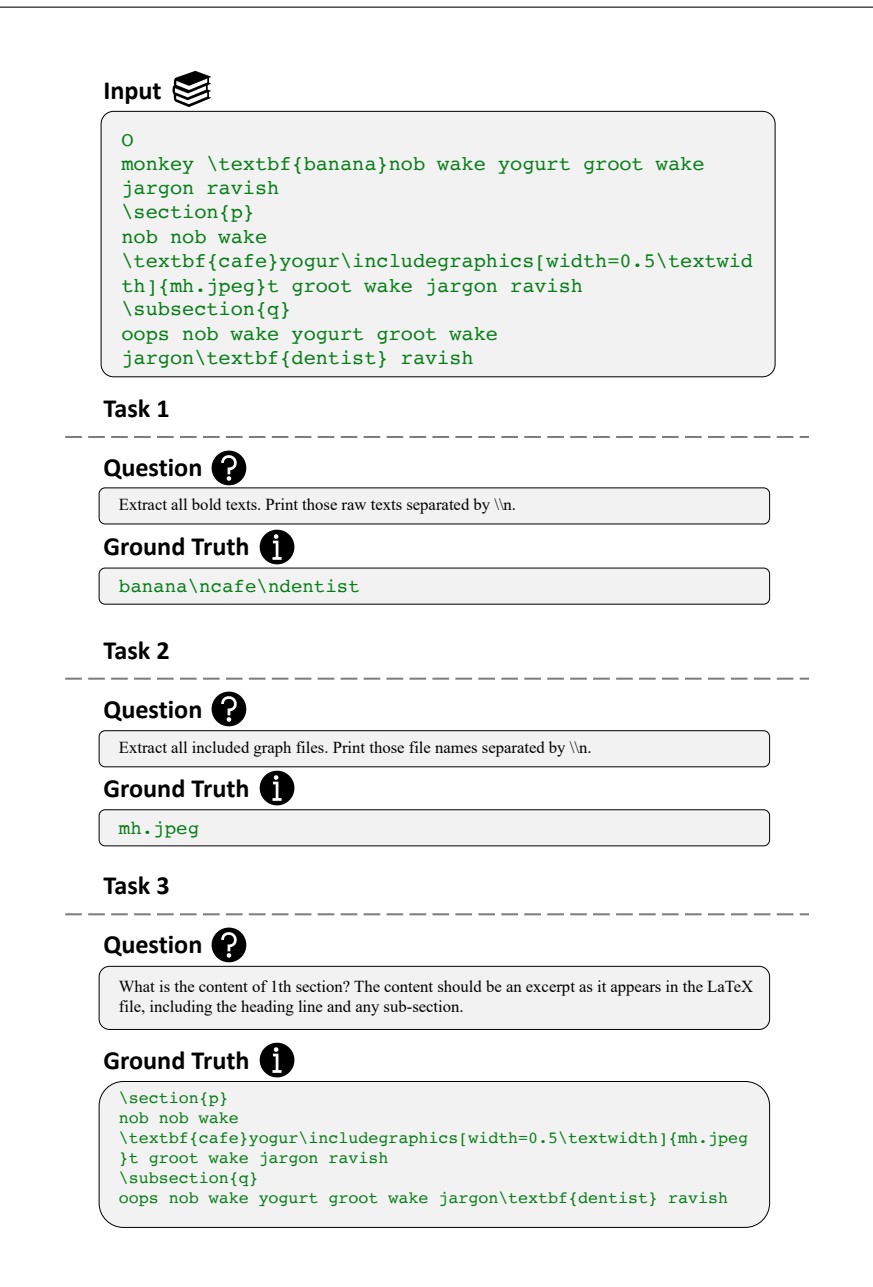

Figure 10: Sample input and tasks of LaTeX.

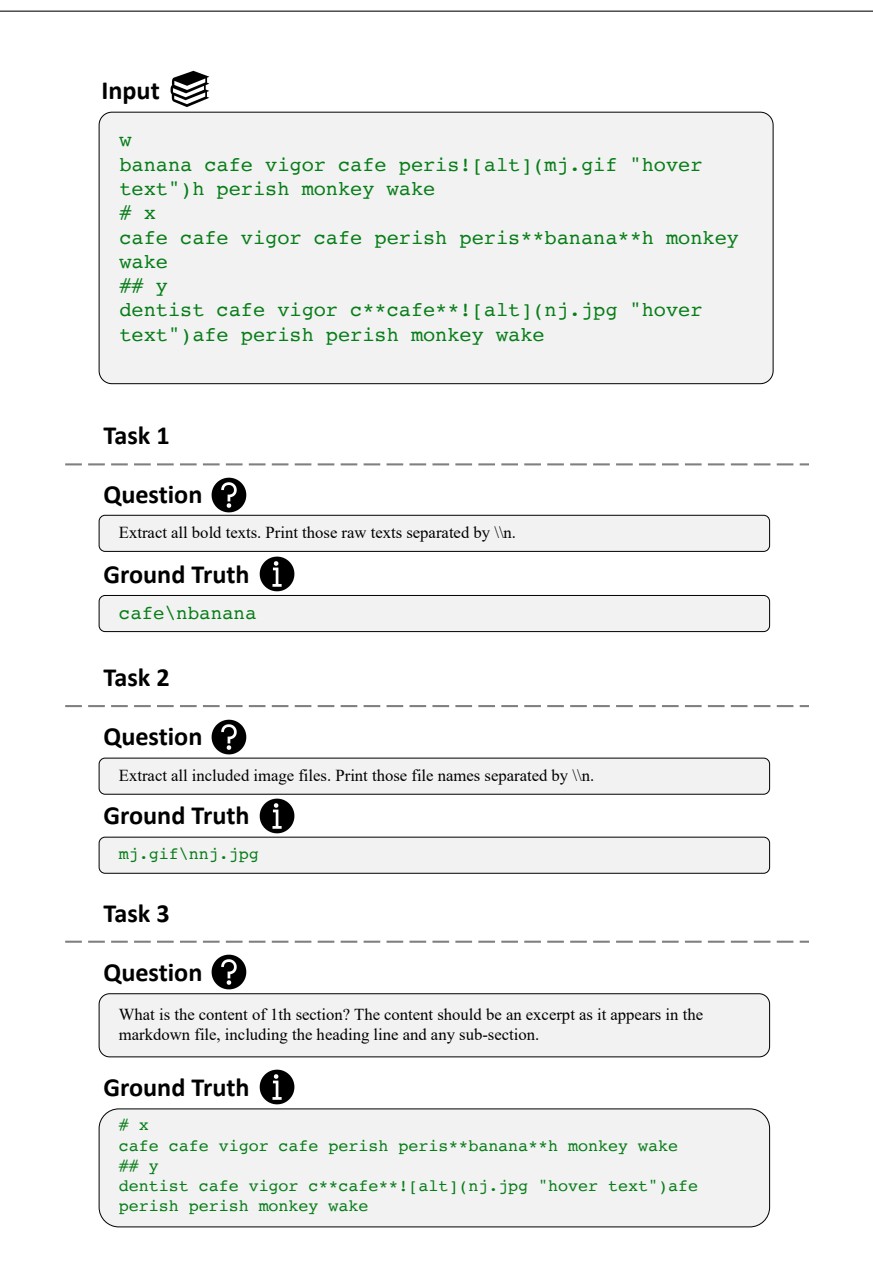

Figure 11: Sample input and tasks of Markdown.

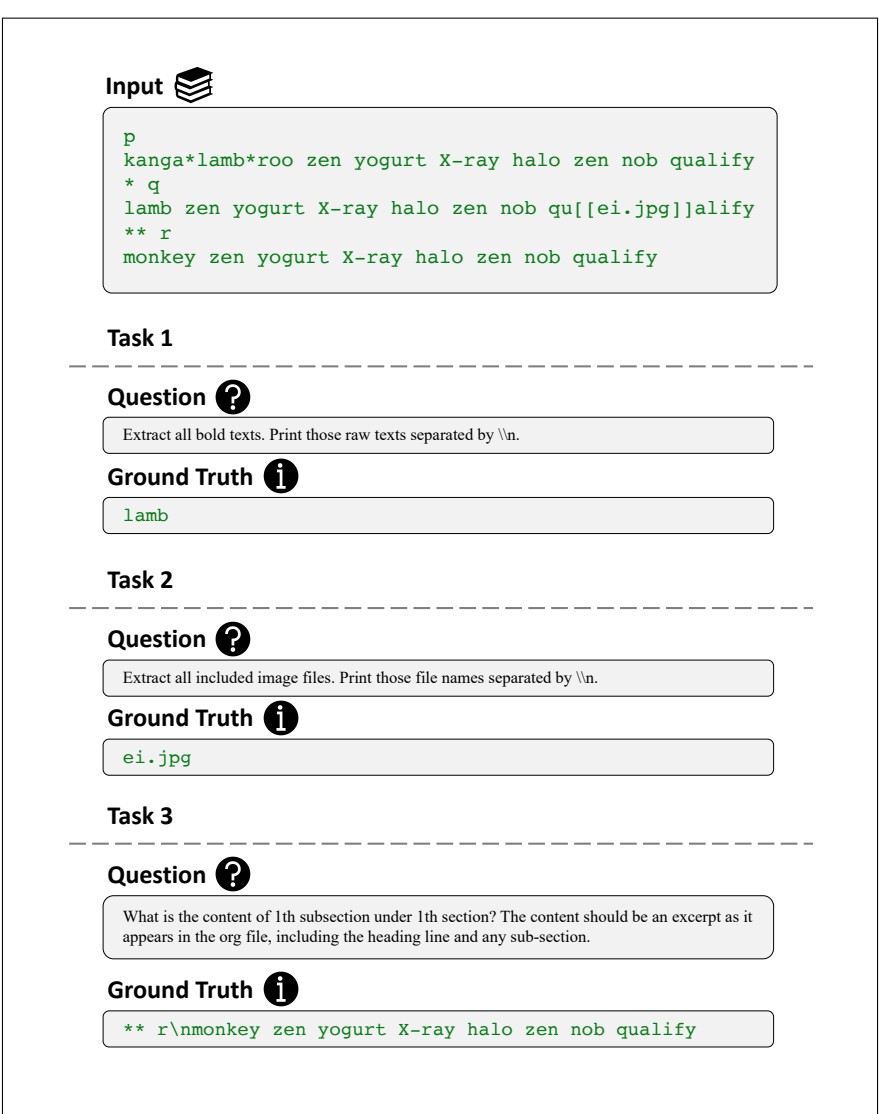

Figure 12: Sample input and tasks of Org.

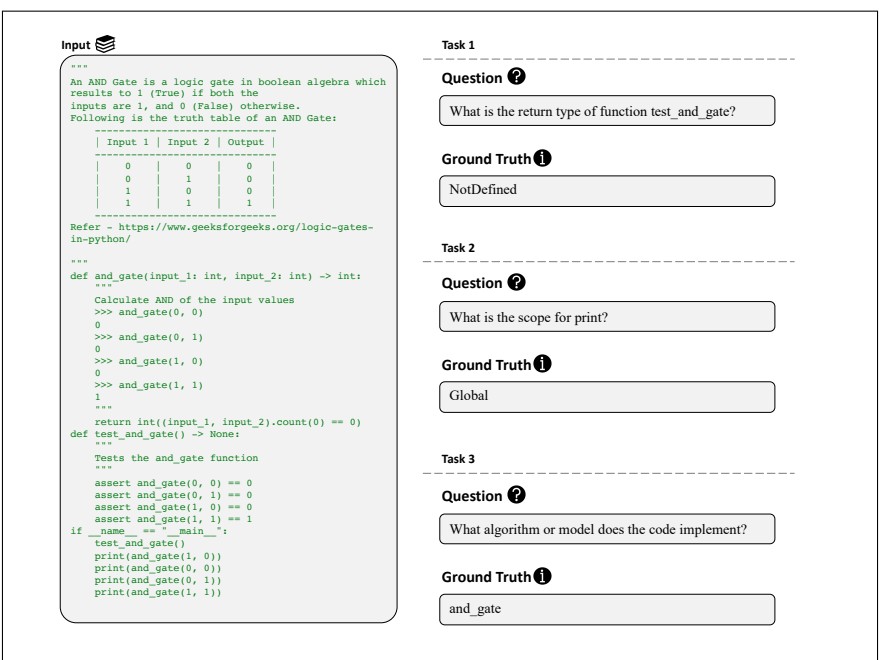

Figure 13: Sample input and tasks of Python codes.

