# OpenReview forum: "Structure-Rich Text Benchmark for Knowledge Inference Evaluation"
_ICLR.cc/2024/Conference — Submitted to ICLR 2024_

### Official Review · Reviewer_k2bc · 2023-10-28

**Soundness:** 1 poor
**Presentation:** 2 fair
**Contribution:** 1 poor
**Rating:** 3
**Confidence:** 4

**Summary:**

This paper constructs a benchmark for LLMs (Large Language Models) composed of structure-rich and syntactically rigorous corpus, in purpose of evaluating the abilities of knowledge inference from small structured text and construction rules.

The contribution of this paper is
1. evaluated the LLMs’ ability to understand and manipulate structure-rich texts
2. present a taxonomy for structured texts and designed structure-related tasks for each class
3. present 32 structure-specific tasks and 2512 samples for our benchmark dataset
4. tested 4 LLMs and present the results under various metrics for assessment

**Strengths:**

1. proposed a benchmark of composed of structure-rich data, such as Tree, Tabular, JSON, YAML, XML, Markdown, Org, LaTeX and PYTHON

**Weaknesses:**

1. The motivation of the paper is not clear. No reference cited in the Background and Motivation section. It seems no previous work focused on this aspect.
2. The related work mentioned in the paper does not justify why understanding the structure-rich data is important.
3. The data construction method is quite simple, just regular expression + simple rules.
4. This paper compares 4 LLM on the proposed benchmark, use GPT4 as baseline and the other 3 models are Minimax, Spark from Xunfei  and Ernie from Baidu. No tables or figures to summarize the experiment results.

**Questions:**

1. Can you list some references about the importance of understanding structure-rich data? What experiments did they do? What are the metrics?
2. What's the novelty in the proposed paper, compared to previous work?

---

### Official Review · Reviewer_qcK5 · 2023-10-30

**Soundness:** 2 fair
**Presentation:** 2 fair
**Contribution:** 1 poor
**Rating:** 1
**Confidence:** 4

**Summary:**

The authors present a new benchmark to probe the abilities of LLMs to generate different types of structured texts including json, xml, yaml, markdown, latex, and the AST from python source code.

**Strengths:**

This paper presents a new benchmark dataset containing ~2500 question answer pairs.

**Weaknesses:**

This work is not very novel and unlikely to be interesting.

**Questions:**

Why wasn't GPT4 used with prompt engineering as well ? Moreover, Table 2, and Table 5 in appendix A.2 suggest that prompt engineering actually decreased performance. Why is that?

---

### Official Review · Reviewer_oBCJ · 2023-10-30

**Soundness:** 2 fair
**Presentation:** 2 fair
**Contribution:** 2 fair
**Rating:** 3
**Confidence:** 3

**Summary:**

This paper proposes a benchmark for evaluating the knowledge inference ability of LLMs.
In particular, the benchmark consists of 2512 QA pairs, covering JSON, YAML, XML, Markdown, Org, LaTeX, Python, etc.
Given the input in one of the aforementioned languages, the LLMs are asked to generated an answer that responds to a paired question in natural language.
The authors conduct experiments using 4 LLMs, and find that current LLMs show poor performance on the benchmark.

**Strengths:**

Comprehensive benchmark that can evaluate LLMs' ability to understand and reason on structured texts.

**Weaknesses:**

1. The experiments and analyses are somehow shallow. In particular, the authors simply compare the performance, while give no discussion. I expect there can be at least more in-depth analysis, for example, more discussion or analysis based on their categories, and what is the possible reason that LLMs fail on those structured texts, and how it can be improved.

2. The presentation is poor and can be further improved. I suggest the authors clearly state what kind of benchmark they are constructing in their introduction - I did not understand it until Section 3, and then realized that it was a QA format. Also, there is no reference in their Section 1, while most statements need supporting evidences/references.

Also, I would like to remind the authors that it is not a proper manner to put most of their key results in the Appendix (and no discussion and analysis is given!) - reviewers are NOT asked to read their appendix.

**Questions:**

Please see weakness

---

### Official Review · Reviewer_yLrc · 2023-11-01

**Soundness:** 3 good
**Presentation:** 3 good
**Contribution:** 3 good
**Rating:** 6
**Confidence:** 3

**Summary:**

This paper curates a benchmark to test LLMs’ capabilities on understanding and generating structurally rich text such as JSON, YAML, Markdown, Python. It first categorize structure-rich text into four representative classes, and then craft specific tasks in each category.

**Strengths:**

- The benchmark is of great practical value because the understanding and generating structure-rich text is very useful  in many real-life scenarios, yet the capability is not evaluated rigorously in the literature.
- The benchmark encompasses a wide range of structure-rich text.

**Weaknesses:**

- The taxonomy that covers widely used structure-rich text is nice, it’d much better if the taxonomy can be further extended to explain the underlying principles for designing specific tasks for each kind of structure-rich text. This would provide insights into what kind of underlying capabilities are being evaluated (e.g., understanding recursive paths should be required in many structure-rich text), and hints on what should/can be improved for future work.
- Only closed-source LLMs are used for evaluation. Results from open-source models should be included for better reproducibility.

**Questions:**

Python files are collected from Internet while files from other text classes are created procedurally (i.e., they're synthetic). I wonder how much does it cost to consider real-world files for all text classes? It seems that most of human effort is needed in curating the tasks given files.

---

### Meta-Review · Area_Chair_WUAf · 2023-12-13

**Metareview:**

The paper presents an automatically acquired benchmark for LLMs for evaluating knowledge inference.

**Justification For Why Not Higher Score:**

Reviewers note shallow experimentation, poor overall presentation.  In general, reviewers are confused.

**Justification For Why Not Lower Score:**

n/a

---

### Decision · Program_Chairs · 2024-01-16

Reject